# Comprehensive Transcriptomic and Metabolic Profiling of *Agrobacterium*-*tumefaciens*-Infected Immature Wheat Embryos

**DOI:** 10.3390/ijms24098449

**Published:** 2023-05-08

**Authors:** Weiwei Wang, Jinliang Guo, Jiayang Ma, Zhulin Wang, Lining Zhang, Zixu Wang, Min Meng, Chao Zhang, Fengli Sun, Yajun Xi

**Affiliations:** State Key Laboratory of Crop Stress Biology for Arid Areas, College of Agronomy, Northwest A&F University, Yangling, Xianyang 712100, Chinaguojinliang@nwafu.edu.cn (J.G.); mjy1270557050@nwafu.edu.cn (J.M.); mengmin@nwsuaf.edu.cn (M.M.);

**Keywords:** wheat (*Triticum aestivum* L.), immature embryo, transcriptome, metabolome, biotic stress

## Abstract

The transformation efficiency (TE) was improved by a series of special chemical and physical methods using immature embryos from the cultivar Fielder, with the PureWheat technique. To analyze the reaction of immature embryos infected, which seemed to provide the necessary by *Agrobacterium tumefaciens* in PureWheat, a combination of scanning electron microscopy (SEM), complete transcriptome analysis, and metabolome analysis was conducted to understand the progress. The results of the SEM analysis revealed that *Agrobacterium tumefaciens* were deposited under the damaged cortex of immature embryos as a result of pretreatment and contacted the receptor cells to improve the TE. Transcriptome analysis indicated that the differentially expressed genes were mainly enriched in phenylpropanoid biosynthesis, starch and sucrose metabolism, plant–pathogen interaction, plant hormone signal transduction, and the MAPK (Mitogen-activated protein kinase) signaling pathway. By analyzing the correlation between differentially expressed genes and metabolites, the expression of many genes and the accumulation of metabolites were changed in glucose metabolism and the TCA cycle (Citrate cycle), as well as the amino acid metabolism; this suggests that the infection of wheat embryos with *Agrobacterium* is an energy-demanding process. The shikimate pathway may act as a hub between glucose metabolism and phenylpropanoid metabolism during *Agrobacterium* infection. The downregulation of the *F5H* gene and upregulation of the *CCR* gene led to the accumulation of lignin precursors through phenylpropanoid metabolism. In addition, several metabolic pathways and oxidases were found to be involved in the infection treatment, including melatonin biosynthesis, benzoxazinoid biosynthesis, betaine biosynthesis, superoxide dismutase, and peroxidase, suggesting that wheat embryos may be under the stress of *Agrobacterium* and, thus, undergo an oxidative stress response. These findings explore the physiological and molecular changes of immature embryos during the co-culture stage of the PureWheat technique and provide insights for *Agrobacterium*-mediated transgenic wheat experiments.

## 1. Introduction

*Agrobacterium*-mediated genetic transformation and biolistic transformation are commonly used in plant systems, among the many methods of delivering DNA into cells [1]. Compared with biolistic particles, *Agrobacterium*-mediated transformation has some unique advantages, including low-copy-number integration, low cost, high stability, and a simple procedure [2,3]. For a long time, the development of transgenic wheat has been slow, and the transformation efficiency (TE) has been at a low level. The first transgenic wheat (*Triticum aestivum* L.) plant was obtained by biolistic particle bombardment in 1992. *Agrobacterium*-mediated transgenic wheat plants were first developed in 1997, and the protocol required about 3 months to obtain stable transgenic plants [4]. After that, the transgenic scheme was continuously optimized; however, the transformation efficiency remained at a low level of 5%. In 2014, Ishida et al. developed the PureWheat technique, in which immature embryos of the cultivar Fielder were used as receptors for *Agrobacterium* infection, greatly improving the transformation efficiency, reaching 50–90% [5]. In this technique, some critical steps were applied carefully, including centrifugation of immature embryos, co-cultivation for two days after *Agrobacterium* infection, recovery after cutting embryo axes, and selection of positive plants. The PureWheat technique sheds light on the genetic engineering of wheat.

Co-cultivation is an important step in the direct contact between *Agrobacterium* and plant receptors, which occurs after *Agrobacterium* infects plant receptors, and generally lasts 1–4 days [6]. Several conditions were proven to be necessary for successful transformation during co-cultivation, including chemicals, actively growing tissues, and temperatures of 22–28 °C [1,6,7]. For example, calli were selected as infection receptors to obtain a high TE in rice and maize, while immature embryos were required in wheat. The transformation ability was enhanced with the addition of cysteine, acetosyringone, and silver nitrate during the co-cultivation process for different plant receptors [8,9]. In conclusion, the co-cultivation stage directly affects the transformation efficiency of *Agrobacterium*-infected plants. However, how the molecular mechanism and endogenous substances change at this stage—which further affects the transformation efficiency—is unknown.

Today, high-throughput omics approaches—including genomics, transcriptomics, proteomics, and metabolomics—have been widely used by researchers to study plants’ life processes and different abiotic stresses, deepening the understanding of multiple biological pathways [10]. Combined analysis of transcriptome RNA sequencing (RNA-Seq) and metabolomics profiling is also increasing. For instance, abscisic acid inhibits embryonic germination [11], leaf, petal, and peel coloring mechanisms [12,13], and differences in carbohydrate and organic acid metabolism between green and red fruits [14]; hydrogen sulfide promotes submergence tolerance [15], root resistance to low nitrogen [16], tolerance mechanisms to cold, salt, drought, and aluminum stress [17,18,19,20,21], an d the response mechanisms to nitrogen, phosphorus, and potassium stress on plant growth [22,23,24,25]. Overall, these multi-omics studies expand our knowledge and understanding of the complex regulatory mechanisms used to respond to various growth processes and stressful environments.

Common wheat is a staple food crop for more than one-third of the world’s population, providing approximately 20% of the daily human calorie consumption. The increasing nutritional demands posed by the growing world population and environmental stresses present major challenges for wheat research and breeders [26,27,28]. In recent years, the widespread application of transgenic technology has provided effective technical support for reducing adverse effects and increasing yields. However, wheat was considered to be a stubborn plant for genetic transformation (due to its low efficiency and genotype dependence) until the development of the PureWheat technique, with a high transformation efficiency. The protocol focuses on the cv. Fielder—a variety that is now widely used in wheat gene transfer—and contains some key steps, such as centrifugation, co-cultivation, embryo axis excision, selection, etc. The process of co-cultivation of *Agrobacterium* and immature embryos is mysterious, including changes in the internal molecular mechanisms caused by the introduction of exogenous genes. In this investigation, we referred to the protocol used in PureWheat and performed an integrated analysis of the transcriptomes and metabolomes of immature embryos from cv. Fielder in the co-culture stage to analyze the changes in gene expression and metabolites during *Agrobacterium* treatment. This helped us to understand the physiological and molecular changes of immature embryos in this regimen after infection by *Agrobacterium tumefaciens*, and it can further explain the mysterious T-DNA insertion of *Agrobacterium tumefaciens* in response to the recipient plant.

## 2. Results

### 2.1. Agrobacterium Adsorbs into Holes on the Surface of Immature Embryos

Ishida et al. developed methods with high transformation efficiency using immature embryos. In this study, wheat embryos from the co-culture stage of the protocol were selected as the research objects to reveal the effects of *Agrobacterium* infection. Firstly, we observed through scanning electron microscopy that the upper epidermis of immature embryos would break and create gaps after the pretreatment (Figure 1a,d). After that, *Agrobacterium* could accumulate in large quantities between the ruptured cell gaps (Figure 1b,e). Interestingly, we found that *Agrobacterium* accumulates little in the smooth cortex but is abundant below the torn cortex (Figure 1c,f). The results showed that *Agrobacterium* accumulates in the cracks produced in pretreated embryos, which should be the initial stage of T-DNA insertion.

Subsequently, some physiological indicators related to oxidative stress were measured (Figure 2). The results showed that the expression of reductases—including superoxide dismutase (SOD) and peroxidase (POD)—was upregulated in the immature embryos treated with *Agrobacterium*, thereby reducing the reactive oxygen species response, which would be conducive to the T-DNA transfer of *Agrobacterium* and the regeneration process of immature embryos. The changes in stress resistance indices included a decrease in glutathione (GSH), an increase in proline, and an increase in Malondialdehyde (MDA), indicating that some stress response occurred in immature embryos.

### 2.2. Transcriptomic Analysis of the Wheat Embryos Infected by Agrobacterium

To gain insight into the molecular events underlying the infection of immature embryos by *Agrobacterium tumefaciens*, whole-genome transcriptomic analysis was employed to analyze the expression of transcriptome genes. Three biological replicates of cDNA libraries were prepared, for a total of four samples of control and treatment after *Agrobacterium* infection of immature embryos. At first, the four cDNA libraries were prepared in Illumina sequencing from three biological replicates per sample, which generated 73,174,653, 75,342,269, 71,580,265, and 72,900,690 sequences with 10.98, 11.3, 10.74, and 10.93 G from CK1, CK2, T1, and T2 respectively (Appendix A).

After the removal of low-quality raw reads—which included those that were empty, too short, or had too many Ns—we obtained an average of 69,898,136, 71,543,825, 65,160,259, and 68,953,701 high-quality sequences for CK1, CK2, T1, and T2, respectively. These reads were assembled into 120,744 total genes and 146,597 total transcripts, with an average length of 1797 bp.

A principal component analysis (PCA) suggested close clustering of replicates within the samples and, hence, excellent sample data reproducibility (Appendix A). A large number of unique sequences from immature embryos should cover a vast majority of the genes in this species.

A total of 1570 DEGs (differentially expressed genes) were identified in the comparison of CK1 vs. T1 (Figure 3a,b, Appendix A); GO (Gene Ontology) annotation of these genes can be divided into molecular function (2019 sequences), biological process (1005 sequences), and cellular components (2072 sequences). A total of 1155 DEGs were identified in the comparison of CK2 vs. T2 (Figure 3b, Appendix A), and the GO annotations of these genes could be classified into molecular functions (1448 sequences), biological processes (512 sequences), and cellular components (2057 sequences). GO term annotation provided a broad overview of the functional groups of genes catalogued in our wheat embryos’ transcriptome. In addition, the Venn diagram showed 184 common upregulated and 271 downregulated differentially expressed genes in CK1 vs. T1 vs. CK2 vs. T2 (Appendix A).

As shown in Figure 3c,d, Kyoto Encyclopedia of Genes and Genomes (KEGG) pathway enrichment analysis revealed that the 124 DEGs in of wheat CK1 vs. T1 were mainly related to phenylpropanoid biosynthesis (162, 6.99%), starch and sucrose metabolism (130, 5.61%), plant–pathogen interaction (108, 4.66%), plant hormone signal transduction (100, 4.31%), the MAPK signaling pathway—plant (82, 3.54%), galactose metabolism (60, 2.59%), flavonoid biosynthesis (58, 2.50%), amino sugar and nucleotide sugar metabolism (58, 2.50%), purine metabolism (49, 2.11%), and pyrimidine metabolism (48, 2.07%), while the 120 DEGs in CK2 vs. T2 were enriched in phenylpropanoid biosynthesis (111, 6.73%), plant–pathogen interaction (104, 6.30%), plant hormone signal transduction (94, 5.70%), starch and sucrose metabolism (83, 5.03%), the MAPK signaling pathway—plant (72, 4.36%), DNA replication (42, 2.55%), protein processing in the endoplasmic reticulum (38, 2.30%), nucleocytoplasmic transport (36, 2.18%), amino sugar and nucleotide sugar metabolism (35, 2.12%), and galactose metabolism (34, 2.06%).

### 2.3. Validation of RNA-Seq Data by Real-Time Quantitative Reverse-Transcription PCR (qRT-PCR)

To determine the accuracy of the RNA sequencing results, nine DEGs from wheat embryos treated with *Agrobacterium tumefaciens* were selected for qRT-PCR (Appendix A). The expression of these genes by qRT-PCR was consistent with the transcriptome sequencing data, validating the usability of the transcriptomic data.

### 2.4. Metabolic Analysis of the Wheat Embryos Infected by Agrobacterium

To fully understand the metabolic changes that occurred in response to *Agrobacterium* stimulation of wheat embryos, a non-target metabolic analysis was performed using UPLC-qTOF-MS; there were 620 known metabolites among the 7174 metabolites detected (Appendix A), and most of the compounds were classified into carboxylic acids and derivatives (88), organooxygen compounds (64), benzene and substituted derivatives (38), fatty acyls (35), cinnamic acids and derivatives (31), phenols (25), indoles and derivatives (16), purine nucleosides (13), imidazopyrimidines (13), keto acids and derivatives (13), steroids and steroid derivatives (13), pyrimidine nucleotides (12), prenol lipids (11), and organonitrogen compounds (10).

Principal component analysis (PCA; Figure 4a,b) showed that the same treatments were gathered together, indicating good repeatability between samples, while different treatments were separated from one another, indicating that there were different effects on metabolites between treatments.

We screened differentially accumulated metabolites (DAMs) with a screening criterion of FC > 2/q < 0.05 and compared the metabolite numbers between the two control conditions and the two treatment levels to identify DAMs following *Agrobacterium* infection of immature embryos. The differentially accumulated metabolites were extracted and placed on a pie chart (Figure 4c). These DAMs mainly included organic acids and derivatives (34), organoheterocyclic compounds (29), organic oxygen compounds (26), benzenoids (19), lipids and lipid-like molecules (17), phenylpropanoids and polyketides (12), nucleosides, nucleotides, and analogues (11), organic nitrogen compounds (4), and alkaloids and derivatives (1). We identified 248 DAMs from CK1 and T1, of which 154 compounds were increased and 94 compounds were decreased (Appendix A). A total of 304 DAMs were identified from CK2 and T2, of which 175 compounds were increased and 129 compounds were decreased (Appendix A). In addition, Venn diagrams were used to identify differential metabolites at the 24 h and 48 h treatments. In total, 92 upregulated common differential metabolites were identified in the comparison of CK1 vs. T1 vs. CK2 vs. T2, and 55 downregulated common differential metabolites were identified in the comparison of CK1 vs. T1 vs. CK2 vs. T2 (Figure 4d,e).

Comparative analysis of immature embryos treated with *Agrobacterium tumefaciens* showed significant differences in metabolites and significant enrichment pathways for CK1 and T1 (49 in total), including purine metabolism, phosphonate and phosphinate metabolism, histidine metabolism, arginine and proline metabolism, tryptophan metabolism, phenylpropanoid biosynthesis, phenylalanine metabolism, alanine, aspartate and glutamate metabolism, and vitamin B6 metabolism. Pathways significantly enriched in CK2 and T2 (a total of 54 pathways) included phenylpropanoid biosynthesis, purine metabolism, tryptophan metabolism, the pentose phosphate pathway, phosphonate and phosphinate metabolism, glutathione metabolism, phenylalanine metabolism, tyrosine metabolism, monoterpenoid biosynthesis, and glycine, serine, and threonine metabolism.

### 2.5. Association Analysis between DEGs and DAMs

To better understand the relationships between genes and metabolites after *Agrobacterium* infects immature embryos, differentially expressed genes and differentially accumulated metabolites were mapped to the KEGG pathway map simultaneously. The results showed that the same pathway of DEGs and DAMs was enriched in glycolysis, the TCA cycle, amino acid biosynthesis, phenylpropionin, lignin, and other related pathways As shown in Figure 5, many genes were changed in *Agrobacterium*-infected immature embryos through glycolysis metabolism, TCA cycle metabolism, and their related pathways; the upregulated genes included *PFK* (*TraesCS3D02G109600*), *gapN* (*TraesCS2D02G197300*), *gpmI* (*TraesCS4B02G172700*), *pyk* (*TraesCS2D02G561700*), *RAFS* (*TraesCS3B02G133400*), *tktA* (*TraesCS2D02G073900*), *IDH1* (*TraesCS2A02G205900*), *GLT1* (*TraesCS3A02G266300*, *TraesCS3B02G299800*, *TraesCS3D02G266400*), and *POP2* (*TraesCS2A02G421400*), and the downregulated genes included *PDHA* (*TraesCS6B02G342000*), *PDHB* (*TraesCS5B02G290600*), *DLAT* (*TraesCS4A02G481800*), *DLD* (*TraesCS1D02G109600*), *ACO* (*TraesCS6B02G342000*), *LSC1* (*TraesCS5B02G290600*), *LSC2* (*TraesCS7D02G011600*), *SDHA* (*TraesCS1D02G109600*), *fumC* (*TraesCS2A02G336500*), *MDH2* (*TraesCS1A02G412900*), and *GSS* (*TraesCS5B02G104500*, *TraesCS7D02G431500*). The upregulation of *PFK*, *gapN*, and *gpmI* promoted the accumulation of 2PGA (2-Phospho-D-glycerate), which promoted the accumulation of quinate and 5-o-(1-carboxyvinyl)-3-phosphoshikimate in the downstream shikimate pathway. *Pyk* was upregulated 2-fold in immature embryos treated with *Agrobacterium tumefaciens*, which is a key enzyme that catalyzes the conversion of PEP (Phosphoenolpyruvate) to pyruvate and then promotes the biosynthesis of the important downstream osmotic substance betaine. Acetyl-CoA (Acetyl coenzyme A) is a hub metabolite linking the glycolytic pathway and the TCA cycle. Several regulated genes were downregulated, including *PDHA*, *PDHB*, *DLAT*, and *DLD*, suggesting that *Agrobacterium* infection activates the glycolytic pathway of immature embryos, and several metabolites enter the shikimic acid pathway through PEP. Six genes encoding five enzymes in the TCA cycle pathway were downregulated in *Agrobacterium*-infected embryos. GSH is an important regulatory metabolite and affects the metabolic process in cells; the GSH-encoding enzyme *GSS* gene was downregulated, and the accumulation of its downstream metabolites in immature embryos was increased. The biosynthesis of GSH was mediated by succinyl-CoA and 2-oxoglutarate in the TCA cycle. Meanwhile, we also detected an accumulation of intermediate metabolites, including γ-glutamylcysteine, glutamate, GABA, and succinate semiphosphate. The results indicated that the TCA cycle pathway participated in the response of immature embryos to *Agrobacterium* and, finally, participated in the GSH biosynthesis pathway through intermediate metabolites. In addition, we also detected changes in some amino acid metabolites, including D-glucuronide, D-serine, L-histidine, isoleucine, D-aspartate, β-alanine, tyrosine, etc.

Structural genes such as *PAL* and *4CL* were involved in the phenylpropanoid metabolism pathway, and many secondary metabolites in this pathway were changed, including decreases in the accumulation of phenylalanine and cinnamic acid, and an increase in p-coumarate accumulation. We found that many enzymes and metabolites were involved in the process of infecting wheat embryos by *Agrobacterium* in related synthetic pathways, such as phenylpropionin and lignin (Figure 6). Structural genes such as *PAL* and *4CL* were involved in the phenylpropanin metabolism pathway, where multiple secondary metabolites were altered, including decreased accumulation of phenylalanine and cinnamic acid, and increased accumulation of p-coumarate. Following the phenylpropyl pathway into the lignin pathway, several enzyme-encoding genes—such as *CCR*, *REF1*, *F5H*, *bglX*—were changed, leading to the accumulation of sinapic acid, cinnamaldehyde, and coniferaldehyde. In addition, some metabolites linked through pathways such as phenylpropylene and lignin were also changed. The downregulation of genes encoding *bx1–5* and upregulation of *bx8–9* genes led to an altered accumulation of melatonin. The expression of these genes reduces the accumulation of DIBOA-glucoside—the key metabolite downstream of this pathway. Benzoic acid is the key substance of adipose biosynthesis, and its accumulation in wheat embryos also increased. In conclusion, immature embryos infected with *Agrobacterium tumefaciens* participate in the phenylpropanoid and lignin metabolic pathways by regulating some key enzymes and leading to changes in metabolites. Additionally, some metabolites related to oxidative stress response are accumulated, indicating that immature embryos express some stress responses through the phenylpropanoid and lignin pathways after *Agrobacterium* stimulation.

## 3. Discussion

Wheat is one of the most important food crops, which is considered to be a recalcitrant plant for genetic transformation due to its low efficiency and genotype dependency until 2014, when Ishida et al. obtained a high efficiency of 40–90% using immature embryos through special treatments. Many previous studies have focused on callus formed from seeds or single-omics analyses of embryos [29]. In this investigation, transcriptome and metabolome analyses were used to analyze the molecular mechanisms of *Agrobacterium*-infected immature embryos, and the changes in their genes and metabolites were analyzed. The citric acid cycle is an important metabolic pathway for the energy supply of organisms, which provides precursors for many biosynthetic pathways through the oxidative breakdown of common oxidative pathways, including carbohydrates, fats, and amino acids [30]. Additionally, citrate-related energy pathways are involved in most callus- or embryo-related studies [31]. Glucose metabolism, flavonoid biosynthesis, and genes related to stress have been shown to be involved in the response of wheat cells to *Agrobacterium tumefaciens*. Many carbohydrate and amino acid metabolites were also found in maize embryos’ callus, induced by the immature embryos [32,33,34]. Genes related to starch/sucrose metabolism were identified in rice and camphor embryos. Furthermore, lignin metabolism occurred in sorghum callus with high regenerative capacity, carbon metabolism, amino acid biosynthesis, glycolysis/gluconeogenesis, etc., identified in somatic maize embryos [34,35,36].

### 3.1. The Damage to Immature Embryos Promoted the Surface Colonization of Agrobacterium

Under natural conditions, *Agrobacterium tumefaciens* lives on the surface of the soil and can perceive host cells through flagella movement or chemotaxis of inducers released from plant tissue wounds, and it temporarily attaches to cells and transmits T-DNA through the formation of solid biofilm tissue [37,38]. Similarly, some studies have shown that pretreatment with drying, heat shock, and ultrasound can promote the invasion of *Agrobacterium*-infected explants [39,40,41]. Centrifuge pretreatment of immature embryos using the PureWheat technique can break the shield of immature embryos and cause some wounds among epidermal cells, which may promote the chemotaxis and quorum-sensing of *Agrobacterium tumefaciens* and activate virulence factors. Previous studies have shown that the formation of wounds encourages *Agrobacterium* to infect plant receptors more easily [42,43]. After 48 h of treatment, *Agrobacterium* could not colonize on the scutellum shield (Figure 1c) but gathered in large numbers at the outlet under the epidermis and formed a biofilm (Figure 1e), which seemed to provide the necessary route for the successful transmission of T-DNA. It was also observed via electron microscopy that the colonized *Agrobacterium* were clustered rather than evenly distributed on the tissue surface (Figure 1b,e), further indicating the importance of surface tissue damage to the successful colonization of *Agrobacterium*.

### 3.2. Energy-Metabolism-Related Pathways Are Involved in the Regulation of Agrobacterium Infection of Immature Embryos

Combined analysis of transcriptomics and metabolomics showed that glycolysis, the TCA cycle, amino acid metabolism, and other related pathways are involved in the molecular regulation mechanisms of wheat embryos after infection with *Agrobacterium*.

Many genes in the TCA cycle were downregulated, including *ACO*, *LSC1*, *LSC2*, *SDHA*, *fumC*, and *MDH2.* Downregulation of *ACO* decreased the accumulation of cis-aconitate in immature embryos. Downregulation of *SDHA* increased the fumarate content. Glutathione (GSH) is a tripeptide composed of glutamic acid, cysteine, and glycine, which is involved in the tricarboxylic acid cycle and glucose metabolism to obtain energy, and is able to participate in the metabolism of products arising from oxidative processes [44]. GSH and glutamate intertransform with one another under the action of two key genes encoding enzymes. The content of glutamate changed through two TCA-derived pathways, which affected the precursors 2-oxoglutarate, succinate semialdehyde, and GABA. Studies have shown that callus growth is a process that requires energy, and the upstream metabolites and regulatory genes of GSH in the study were changed, which suggested that GSH was involved in the oxidative process of *Agrobacterium* infection of immature embryos through the TCA cycle.

Meantime, many genes and metabolites were also altered in the glycolytic pathway connected by the TCA cycle. The upregulation of *PKM*, *gapN*, and *gpmI* led to the accumulation of the important intermediate 2PGA, while the upregulation of pyk reduced the accumulation of pyruvate—an end-product of the glycolytic pathway. Acetyl-CoA is the starting substrate of the tricarboxylic acid cycle and is an intermediate of sugar metabolism, as well as a metabolite of fats and certain amino acids. The genes encoding the enzyme of acetyl-CoA were downregulated, including *PDHA*, *PDHB*, *DLAT*, and *DLD*. The shikimate pathway starts from the pep of the glycolytic pathway, which leads to the biosynthesis of aromatic amino acids that are essential for protein biosynthesis and the production of a wide array of secondary plant metabolites. Among them, quinate is an astringent feeding deterrent that can be formed in a single-step reaction from 3-dehydroquinate, catalyzed by quinate dehydrogenase (QDH) [45]. 5-O-(1-carboxyvinyl)-3-phosphoshikimate, an important metabolite of the shikimate pathway, accumulated in *Agrobacterium*-treated immature embryos. It has been hypothesized that the accumulation of quinate may be involved in the metabolic pathway of shikimate, affecting the synthesis of alanine (Phe), tyrosine (Tyr), and tryptophan (Trp). In addition, many amino acids were detected to be involved in the metabolic activities of the infected immature embryos through the TCA cycle and the glycolytic pathway, including D-glucuronide, D-serine, L-histidine, isoleucine, D-aspartate, and β-alanine. Numerous studies have shown that actively growing tissues are involved in pathways related to energy metabolism, including the TCA cycle, glycolysis, amino acid metabolism, etc. Immature embryos stimulated with *Agrobacterium* were also involved in these processes, as in previous descriptions.

### 3.3. Wheat Embryos Infected by Agrobacterium tumefaciens Participated in Oxidative Stress and Biological Stress through the Phenylpropanoid Pathway

Phenylpropanoid-based polymers contribute to plants’ responses to abiotic and biotic stresses and are essential for plant development and survival [46,47]. The first three steps of the phenylpropanoid pathway constitute the general phenylpropanoid pathway, which is associated with core and specialized metabolism and provides precursors for downstream metabolites. The reactions in the general phenylpropanoid pathway are catalyzed by phenylalanine ammonia lyase (PAL), cinnamic acid 4-hydroxylase (C4H), and 4-coumarate CoA ligase (4CL). PAL is a gateway enzyme of the general phenylpropanoid pathway, which directs the metabolic flux of the shikimate pathway to various branches of phenylpropanoid metabolism by catalyzing phenylalanine to form trans-cinnamic acid [48,49].

In the phenylpropanoid metabolic pathway, we detected changes in two key enzymes (PAL and 4CL), which affected several metabolic pathways, including lignin biosynthesis and benzoxazinoid biosynthesis.

The upregulation of several genes increases the accumulation of 2PGA in the glycolytic pathway, which prompts the accumulation of 5-o-(1-carboxyvinyl)-3-phosphoshikimate, a key metabolite in the shikimic acid pathway. Moreover, most genes in the TCA cycle linked by acetyl-CoA were downregulated in the experiment. We speculated that the immature embryos treated with *Agrobacterium tumefaciens* did not choose to enter the TCA cycle after the glycolytic pathway was decomposed but entered the phenylpropanin pathway through the shikimic acid pathway, resulting in changes in the metabolites.

The lignin pathway is one of the branches of phenylpropanoid metabolism downstream of the general phenylpropyl pathway. In this study, the downregulated expression of the *F5H* gene led to increased accumulation of 5-hydroxyferulic acid and sinapic acid, while the upregulated expression of the *CCR* gene led to increased accumulation of coniferaldehyde, and aldehyde accumulated under 24 h treatment. These metabolites are important precursors of p-hydroxyphenyl (H), guaiacyl (G), and syringyl (S). Scanning electron microscopy showed that the surface of the immature embryos was damaged after centrifugation, and *Agrobacterium* could enter the intercellular space through these damaged holes and directly contact the cell wall. We hypothesized that the cell wall of immature embryos would be attacked by *Agrobacterium*, which would affect the response mechanism of lignin in the cell wall. Unfortunately, no changes in lignin were detected in this study, which still needs to be confirmed by further experiments.

Melatonin is ubiquitous in plants and promotes tolerance to adverse environmental conditions [50]. In addition, melatonin interacts with plant hormones and regulates gene expression to improve plant growth and stress tolerance [51]. Melatonin can activate downstream stress response pathways to improve tolerance to abiotic and biotic stresses, such as carbohydrate catabolism and antioxidant reactions [52]. The high expression of *ASMT* in *Arabidopsis* leads to an increase in melatonin content [53]. We found that the accumulation of melatonin was altered, and the expression of its enzyme-encoding gene *ASMT* was significantly increased. The results suggest that melatonin is involved in the response of immature embryos to *Agrobacterium tumefaciens*. Moreover, transcriptome analysis revealed that a variety of plant hormones were involved in this process, and changes in POD, SOD, and other oxidases were detected, further indicating that *Agrobacterium* infection caused biological stress on immature embryos, inducing the occurrence of the melatonin synthesis pathway. In addition, we also detected a large accumulation of benzoic acid in immature embryos, and the functional groups of benzoic acid may induce tolerance to multiple stresses in plants. Combined with lignin biosynthesis, melatonin biosynthesis, benzoxazinoid biosynthesis, and benzoic acid biosynthesis, we used the phenylpropanin synthesis pathway as the pivot pathway to analyze the wheat embryos infected by *Agrobacterium*. A large amount of evidence suggests that wheat embryos will respond to oxidative stress, biological stress, and other related pathways after infection by *Agrobacterium*.

## 4. Materials and Methods

### 4.1. The Cultivation and Culture of cv. Fielder and the Acquisition of Immature Embryos

The seeds of cv. Fielder and the *Agrobacterium* EHA105 strain were preserved in our laboratory. Seeds were cultivated in pots (H21 cm × D21 cm, six plants per pot) containing field-collected soil in a greenhouse with a relative humidity of 70 ± 5%, at 20 ± 2 °C and 16 ± 2 °C during the day and night, respectively. Moreover, air conditioning and supplementary lights were needed to ensure stable growing conditions. Good immature embryos of 2.0 and 2.5 mm in length along the axis were collected at the developmental stage from panicles about 14 days after anthesis, on an extremely clean bench. The infection with *Agrobacterium tumefaciens* was carried out in strict accordance with the protocol provided by Ishida et al. (Appendix A). Samples of immature embryos were collected at 24 h (T1) and 48 h (T2) after *Agrobacterium* infection (Appendix A); immature embryos treated with infectious fluid were selected as controls (CK1, CK2), immediately frozen in liquid nitrogen, and then stored at −80 °C for subsequent transcriptome sequencing and metabolite extraction. Transcriptome sequencing was performed in three independent biological replicates. Metabolites were subjected to six independent biological replicates. EHA105 that carried pTCK303, containing the intron-gus gene and a hygromycin resistance gene, was used in our tests.

### 4.2. SEM Observation and Oxidation Index Determination

To observe the adsorption status of *Agrobacterium* on the surface of immature embryos, the surface information of immature embryos was recorded using a scanning electron microscope (Nova Nano SEM 450, FEI, Hillsboro, OR, USA) after gold-spraying treatment [38].

Physiological indicators were measured with reference to previous studies, and the data were collected and analyzed [54,55,56,57,58,59,60].

### 4.3. RNA Extraction, Quality Control, and RNA-Seq

Total RNA was extracted using TRIzol reagent (Takara, Dalian, China) following the manufacturer’s procedure. The total RNA quantity and purity were analyzed using a Bioanalyzer 2100 and the RNA 6000 Nano LabChip Kit (Agilent, Santa Clara, CA, USA), and high-quality RNA samples with RIN number > 7.0 were used to construct the sequencing library. After total RNA was extracted, mRNA was purified from total RNA (5 μg) using Dynabeads Oligo (dT) (Thermo Fisher, Waltham, MA, USA), with two rounds of purification. Following purification, the mRNA was fragmented into short fragments using divalent cations under elevated temperature. Then, the cleaved RNA fragments were reverse-transcribed to create the cDNA using SuperScript™ II Reverse Transcriptase (Invitrogen, Waltham, MA, USA), which was then used to synthesize U-labeled second-stranded DNAs with *E. coli* DNA polymerase I (NEB, Beverly, MA, USA), RNase H (NEB, Beverly, MA, USA), and dUTP solution (Thermo Fisher, Waltham, MA, USA). An A base was then added, the sequencing connector was connected, and the fragments were purified using the AMPure XP beads to select cDNA fragments of 300 bp in length. Finally, we performed the 2 × 150 bp paired-end sequencing (PE150) on an Illumina Novaseq™ 6000 (LC-Bio Technology Co., Ltd., Hangzhou, China), following the vendor’s recommended protocol. Reads obtained from the sequencing machines including raw reads containing adapters or low-quality bases were removed before downstream analyses.

### 4.4. Transcriptome Data Analysis

The filtered reads were mapped to the wheat reference genome using the HISAT2 (https://daehwankimlab.github.io/hisat2/, version:hisat2-2.2.1, accessed on 10 August 2022) package [61]. FPKM (fragments per kilobase million, normalized based on the count of original reads of genes) was used as a measure of gene expression. StringTie and ballgown (http://www.bioconductor.org/packages/release/bioc/html/ballgown.html, accessed on 10 August 2022) were used to estimate the expression levels of all transcripts and determine the expression abundance for mRNAs by calculating FPKM (fragments per kilobase of transcript per million mapped reads) values [61,62]. Genes’ differential expression analysis was performed using DESeq2 software (v1.22.1) between two different groups [63,64]. The genes with a false discovery rate (FDR) below 0.05 and absolute fold change ≥ 2 were considered to be differentially expressed genes. Differentially expressed genes were then subjected to enrichment analysis of GO functions and KEGG pathways [65,66].

### 4.5. Validation of Transcriptomic Data by qRT-PCR

To verify the accuracy of transcriptome data, quantitative reverse-transcription PCR was performed on 9 DEGs altered after *Agrobacterium* infection. The specific primers were designed using Primer 6.0 (Appendix A). First-strand cDNA was synthesized from 0.5 μg of total RNA, as mentioned above, using the PrimeScript^TM^ RT reagent kit with gDNA Eraser (Takara, Dalian, China), according to the manufacturer’ s instructions.

The synthesis of first-strand cDNA was performed using the PrimeScript^TM^ RT reagent kit with gDNA Eraser (Takara, Dalian, China) according to the manufacturer’ s instructions. Quantitative reverse-transcription PCR was conducted using the QuantStudio 3 Real-Time PCR system (Thermo Fisher, Waltham, MA, USA) with TB Green™ Premix Ex Taq™ II (TaKaRa, Dalian, China). The programmer performed the following steps: 10 min at 95 °C, followed by 40 cycles of 5 s at 95 °C and 31 s at 55 °C. The actin gene was employed as the reference gene for the experiments (Appendix A). All of the samples were examined in triplicate, with two technical replicates. The relative expression of these genes was calculated by the 2^−ΔΔCt^ method [67]. Subsequently, the gene expression data were graphically screened on the bar chart with Microsoft Excel 2016.

### 4.6. Metabolite Extraction and Metabolic Profiling Analysis

The untargeted metabolome was used to analyze the metabolic changes of immature embryos treated with *Agrobacterium*. At first, the collected samples were thawed on ice, and metabolites were extracted with 50% methanol buffer. Briefly, 20 μL of sample was extracted with 120 μL of precooled 50% methanol, vortexed for 1 min, and incubated at room temperature for 10 min; the extraction mixture was then stored overnight at −20 °C. After centrifugation at 4000× *g* for 20 min, the supernatants were transferred into new 96-well plates. The samples were stored at −80 °C prior to the LC-MS analysis. In addition, pooled QC samples were also prepared by combining 10 μL of each extraction mixture [68].

All samples were acquired by the LC-MS system following machine orders. Firstly, all chromatographic separations were performed using a Thermo Scientific UltiMate 3000 HPLC. An ACQUITY UPLC BEH C18 column (100 mm × 2.1 mm, 1.8 μm, Waters, UK) was used for the reversed-phase separation. The column oven was maintained at 35 °C. The flow rate was 0.4 mL/min, and the mobile phase consisted of solvent A (water, 0.1% formic acid) and solvent B (acetonitrile, 0.1% formic acid). The gradient elution conditions were set as follows: 0~0.5 min, 5% B; 0.5~7 min, 5% to 100% B; 7~8 min, 100% B; 8~8.1 min, 100% to 5% B; 8.1~10 min, 5%B. The injection volume for each sample was 4 μL.

The acquired MS data pretreatments—including peak picking, peak grouping, retention time correction, second peak grouping, and annotation of isotopes and adducts—were performed using XCMS software (v3.22.0). Raw LC−MS data files were converted into mzXML format and then processed by the XCMS, CAMERA, and metaX toolbox implemented with R software (v4.2). The intensities of each peak were recorded, and a three-dimensional matrix containing arbitrarily assigned peak indices, sample names, and ion intensity information was generated. The online KEGG HMDB database was used to annotate the metabolites by matching the exact molecular mass data (*m*/*z*) of samples with those from the database. The metaX software (http://metax.genomics.cn) was used to quantify and screen the differential metabolites. Principal component analysis (PCA) was used to normalize the metabolite profiling data. The differential metabolites were detected based on the following parameters: metabolites with VIP ≥ 1 and a fold change ≥ 2 or fold change ≤ 0.5. The different metabolites were subjected to data normalization.

### 4.7. Combined Transcriptome and Metabolome Analyses

A joint analysis of DEGs and DAMs was used to determine the degree of enrichment of the pathways. Pearson’s correlation tests with a PCC > 0.8 were used to examine the associations between gene expression and metabolite content, and the annotated metabolites and genes were then mapped to the KEGG pathways.

### 4.8. Statistical Analysis

SPSS Statistics 22.0 software was used for the data analysis. The difference between the two treatments was determined by one-way analysis of variance and Duncan’s multiple range test at the 0.05 and 0.01 significance levels. The data were displayed using Microsoft Excel 2016.

## 5. Conclusions

This investigation explains the response of immature wheat embryos to *Agrobacterium* infection. Our results showed that immature embryos infected by *Agrobacterium* mainly activated energy- and stress-related signaling pathways, and in this process, redox substances such as SOD, POD, CAT, GSH, proline, and MDA were changed. We also identified several genes that play key roles in this process, including *F5H* and *CCR*. Overall, integrated transcriptomic and metabolomic analyses revealed that immature embryos respond to *Agrobacterium* invasion through changes in energy-metabolism-related pathways, shikimic acid pathways, phenazine synthesis pathways, and phenylpropanoid metabolic pathways; this is a complex signaling regulation process, and further studies should focus on how these candidate genes or metabolites are involved in the process.

## Figures and Tables

**Figure 1 ijms-24-08449-f001:**
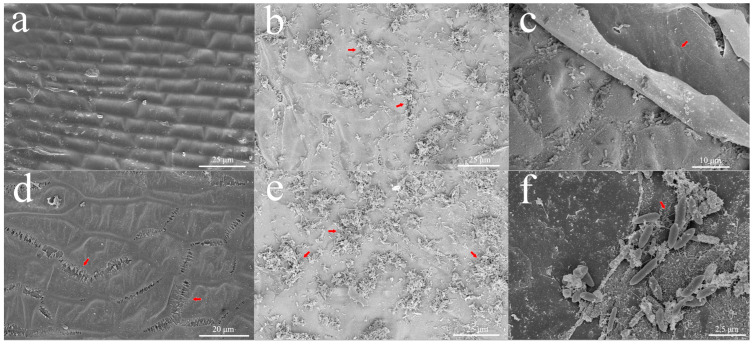
Scanning electron microscopy of the shield of immature embryos showed that *Agrobacterium* cells clustered on the surface of pretreated cracks. Note: (**a**) Wheat embryos without pretreatment treatment have a complete surface. (**b**) Immature embryos treated with *Agrobacterium* for 24 h; the red marks indicate that *Agrobacterium* is clustered in the gap. (**c**) Treated with *Agrobacterium* for 48 h; there were *Agrobacterium* attached under the cortex of immature embryos, and the cortex was smooth, without obvious bacterial attachment; the red mark indicates the smooth upper epidermis. (**d**) A 48 h pretreated (CK2) embryo that was not treated with *Agrobacterium*; the red marks indicate that the smooth upper epidermis of the pretreated wheat embryos breaks and produces cracks. (**e**) Immature embryos treated with *Agrobacterium* for 48 h; the red marks indicate that *Agrobacterium* is clustered in the gap. (**f**) The attachment of immature embryos without a cortex after treatment with *Agrobacterium* for 48 h; The red marks indicate that *Agrobacterium* accumulates under the torn cortex.

**Figure 2 ijms-24-08449-f002:**
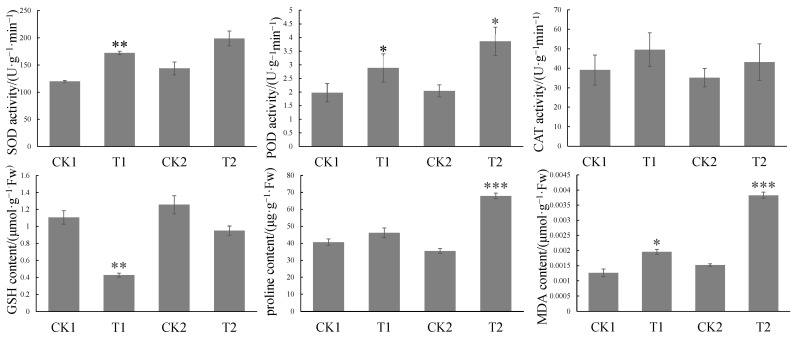
Changes in oxidative-stress-related indices of immature embryos treated with *Agrobacterium* for 24 h and 48 h. Note: CK1 and CK2 refer to immature wheat embryos that were not infected by *Agrobacterium* after 24 h and 48 h of treatment, respectively; T1 and T2 refer to immature wheat embryos that were infected by *Agrobacterium* after 24 h and 48 h of treatment, respectively; * indicates significant difference at *p* < 0.05, ** indicates significant difference at *p* < 0.01, and *** indicates significant difference at *p* < 0.001.

**Figure 3 ijms-24-08449-f003:**
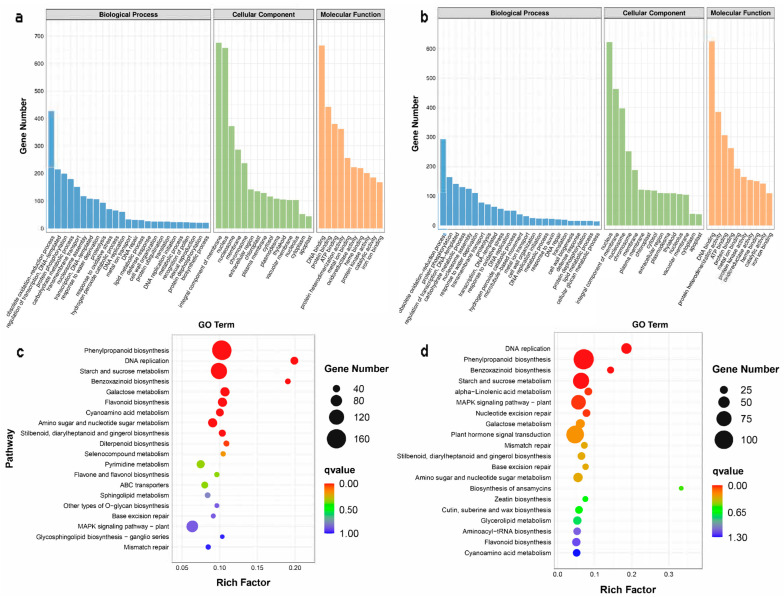
GO classification and KEGG pathways of transcriptomic data of immature wheat embryos infected with *Agrobacterium.* Note: (**a**) GO classification of DEGs in CK1 vs. T1. The *x*-axis represents GO terms, including biological process, cellular component, and molecular function, and the *y*-axis represents the number of DEGs. (**b**) GO classification of DEGs in CK2 vs. T2. The *x*-axis represents GO terms, including biological process, cellular component, and molecular function, and the *y*-axis represents the number of DEGs. (**c**) Scatterplot of KEGG pathways of DEGs in CK1 vs. T1; the *x*-axis represents the degree of enrichment, as shown by Rich factor, *q*-value, and the number of genes enriched in each pathway, while the *y*-axis represents the KEGG terms. (**d**) Scatterplot of KEGG pathways of DEGs in CK2 vs. T2; the *x*-axis represents the degree of enrichment, as shown by Rich factor, *q*-value, and the number of genes enriched in each pathway, while the *y*-axis represents the KEGG terms.

**Figure 4 ijms-24-08449-f004:**
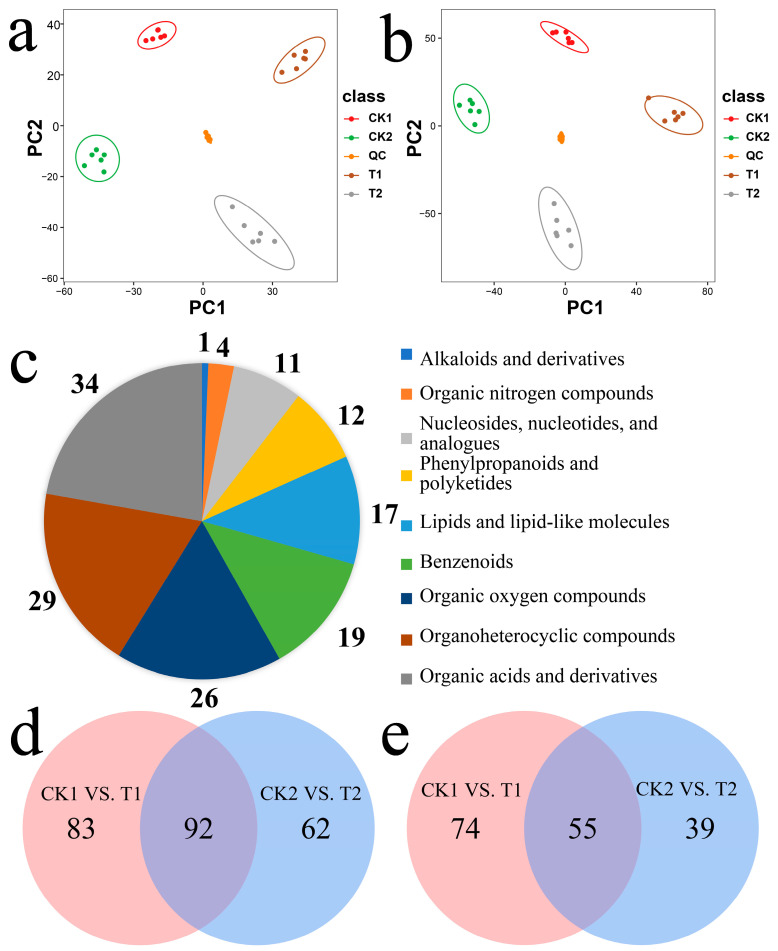
Analysis of differentially accumulated metabolites in immature wheat embryos infected with *Agrobacterium*. Note: (**a**) PCA analysis in samples of CK1 vs. T1. (**b**) PCA analysis in samples of CK2 vs. T2. (**c**) Classification of the isolated compounds in immature embryos infected with *Agrobacterium*. (**d**) Venn diagram showing the differential expression of upregulated metabolites at 24 h and 48 h. (**e**) Venn diagram showing the differential expression of downregulated metabolites at 24 h and 48 h.

**Figure 5 ijms-24-08449-f005:**
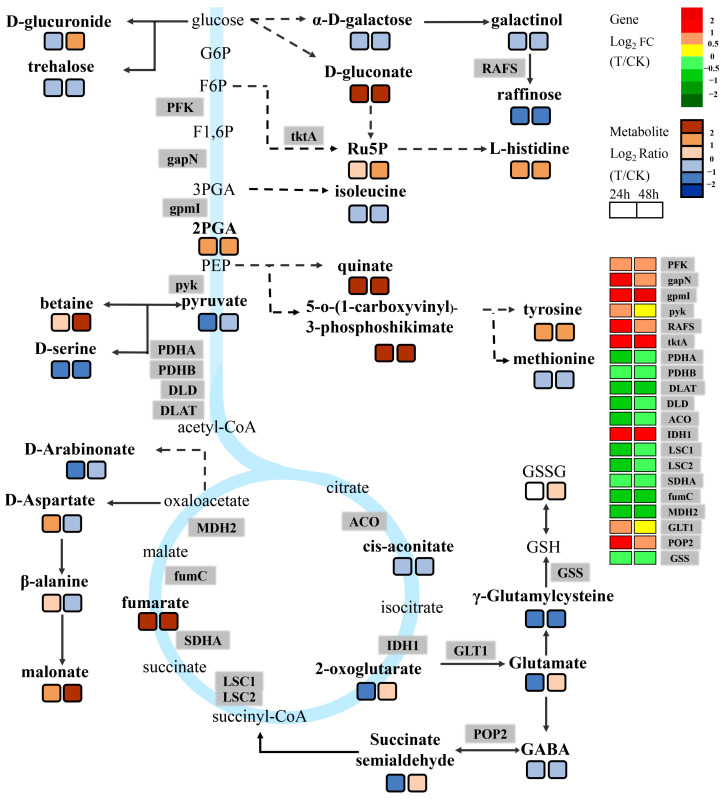
The DEGs and DAMs simultaneously mapped to the carbohydrate metabolism. Note: The grey shading font represents differentially expressed genes, and the detailed changes in expression multiples are indicated by different-colored boxes on the right-hand side of the figure. The brown and blue rectangles represent multiples of differentially expressed metabolites. The rectangles on the left represent the fold changes in gene expression or metabolite accumulation multiples of T2 and CK2; the rectangles on the right represent the fold changes in gene expression or metabolite accumulation multiples of T2 and CK2.

**Figure 6 ijms-24-08449-f006:**
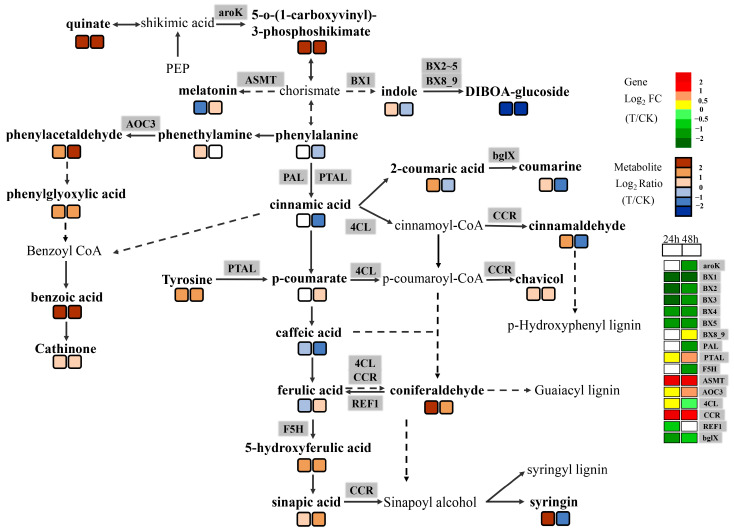
The DEGs and DAMs simultaneously mapped to the monolignol biosynthesis and related oxidative stress pathways. Note: The grey shading font represents differentially expressed genes, and the detailed changes in expression multiples are indicated by different-colored boxes on the right-hand side of the figure. The brown and blue rectangles represent multiples of differentially expressed metabolites. The rectangles on the left represent the fold changes in gene expression or metabolite accumulation multiples of T2 and CK2; the rectangles on the right represent the fold changes in gene expression or metabolite accumulation multiples of T2 and CK2.

## Data Availability

Not applicable.

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
