# Peer review of "Comprehensive Transcriptomic and Metabolic Profiling of *Agrobacterium*-*tumefaciens*-Infected Immature Wheat Embryos"

_ijms, 2023, doi:10.3390/ijms24098449_

Round 1

Reviewer 1 Report

The manuscript ID ijms-2365785 describes the combination of scanning electron microscopy (SEM), complete transcriptome analysis, and metabolome analysis was employed to analyze and understand the reaction progress of wheat immature embryos infected by Agrobacterium tumefaciens in PureWheat. The manuscript is very interesting and involves important information for readership. However, some issues should be addressed before being considered further.

1.      Line 9: Place the TE abbreviation correctly.

2.      Figure 1: Improve the caption information to become self-explanatory.

3.      Figures 2, 3, 4, 5, 6. Improve the quality of this figure since it is challenging to visualize the information. In addition, a self-explanatory caption should also be provided. Each subfigure and the main elements must be explained in the caption.

4.      Line 501: The sequences of the specific primers and the housekeeping genes must be informed.

5.      Line 450: The origin of the Agrobacterium strain and the cv. Fielder must be informed.

6.      Line 453: Details on experimental design should be provided.

7.      Line 532: More details and parameters used for LC-MS data preprocessing must be informed.

8.      The conclusion section can be improved since there is a summary of results, but conceptual findings from the mechanistic point of view should be provided (e.g., from Figures 5 and 6).

9.      Supplementary material is missing but mentioned in the manuscript.

10. Line 386: "phenylpropanoid" instead of "phenylpropane"

11. Figure 2: Statistical analysis must be performed over these data to explain significant differences.

Detailed scrutiny should be performed throughout the manuscript to look for some grammar, stylistic, and even typos issues.

Author Response

Dear Editors and Reviewers

Thank you for your letter and for the reviewers’ comments concerning our manuscript entitled Comprehensive Transcriptome and Metabolic Profiling of Agrobacterium tumefaciens Infected immature Embryos in Wheat.

Those comments are all valuable and very helpful for revising and improving our paper, as well as the important guiding significance to our researches. We have studied comments carefully and have made correction which we hope to meet with approval. Revised portion are marked in the paper. The main corrections in the paper and the responds to the reviewer’s comments are as flowing:

  1. Line 9: Place the TE abbreviation correctly.

ANSWERS: The TE abbreviation placed correctly.

  1. Figure 1: Improve the caption information to become self-explanatory.

ANSWERS: It has been improved in the article.

  1. Figures 2, 3, 4, 5, 6. Improve the quality of this figure since it is challenging to visualize the information. In addition, a self-explanatory caption should also be provided. Each subfigure and the main elements must be explained in the caption.

ANSWERS: We have re-inserted high-quality clear images into the manuscript, as well as providing additional PDF files. We checked all the captions and made it became more compatible with the images. In addition, the unknown elements in the figures are explained in detail in the note.

  1. Line 501: The sequences of the specific primers and the housekeeping genes must be informed.

ANSWERS: The sequences of the specific primers and the housekeeping genes were informed in supplemental table 5.

  1. Line 450: The origin of the Agrobacterium strain and the cv. Fielder must be informed.

ANSWERS: The seeds of cv. Fielder and the Agrobacterium EHA105 strain were preserved in our laboratory. It has been added in materials and methods.

  1. Line 453: Details on experimental design should be provided.

ANSWERS: Supplementary file 3 provides the details on experimental design.

  1. Line 532: More details and parameters used for LC-MS data preprocessing must be informed.

ANSWERS: We have perfected LC-MS data preprocessing in the manuscript

  1. The conclusion section can be improved since there is a summary of results, but conceptual findings from the mechanistic point of view should be provided (e.g., from Figures 5 and 6).

ANSWERS: According to the results of the experiment, we reimagined the conclusion, the integrated transcriptomic and metabolomic analyses revealed that wheat immature embryos respond to Agrobacterium invasion through changes in energy metabolism-related pathways, shikimic acid pathways, phenazine synthesis pathways, and phenylpropanoid metabolic pathways. and we also identified several genes that play key roles in this process, including F5H, CCR, PAL 4CL, etc. At present, there are few integrated analyses of the transcriptome and metabolome of agrobacterium treatment. With further research, more mechanisms of Agrobacterium insertion will be revealed.

  1. Supplementary material is missing but mentioned in the manuscript.

ANSWERS: Supplementary materials have been submitted the system together with the manuscript.

  1. Line 386: "phenylpropanoid" instead of "phenylpropane"

ANSWERS: Mistakes in the manuscript have been modified.

  1. Figure 2: Statistical analysis must be performed over these data to explain significant differences.

ANSWERS: All data in the manuscript were statistically analyzed, and the significance of the differences was explained around the chart.

We tried our best to improve the manuscript and made some changes in the manuscript. These changes will not influence the content and framework of the paper. And here we did not list the changes but marked in red in revised paper.

We appreciate for Editors/Reviewers’ warm work earnestly and hope that the correction will meet with approval.

Once again, thank you very much for your comments and suggestions

Sincerely

Yajun Xi

30 April 2023

Reviewer 2 Report

The article is a high-level work and the authors obtained significant very interesting results, which will undoubtedly make a great contribution to understanding the mechanisms of agrobacterial transformation of wheat. The article is very interesting, original, the content of the article corresponds to the abstract and title.
Thus the work of authors is definitely relevant and undoubtedly important.

There are some comments and suggestions for authors.
1. Please correct the Figure 2, 3 and 4 (the font is very small and hard to read). Unfortunately I didn't get a chance to see the Supplementry figures but please air them too. Figures should not cause difficulties for readers of your article.
2. In the materials and methods section, it is desirable to describe in detail the steps of the protocol by Ishida et al., it may be worth moving it to the Supplementry materials.
3. And it is better to describe in more detail the process of selecting controls CK1 and CK2. Were they also incubated on WLS-AS medium only without Аgrobacterium tumefaciens? After all, the composition of the medium can affect the results of your research.
How many immature wheat embryos were used?
Since you have a complex study with a large amount of data, it is better to describe the experiment itself in more detail.

The authors have undoubtedly obtained important results, and I believe that the article can be published after revision.

Author Response

Dear Editors and Reviewers

Thank you for your letter and for the reviewers’ comments concerning our manuscript entitled Comprehensive Transcriptome and Metabolic Profiling of Agrobacterium tumefaciens Infected immature Embryos in Wheat.

Those comments are all valuable and very helpful for revising and improving our paper, as well as the important guiding significance to our researches. We have studied comments carefully and have made correction which we hope to meet with approval. Revised portion are marked in the paper. The main corrections in the paper and the responds to the reviewer’s comments are as flowing:

  1. Please correct the Figure 2, 3 and 4 (the font is very small and hard to read). Unfortunately, I didn't get a chance to see the Supplementry figures but please air them too. Figures should not cause difficulties for readers of your article.

ANSWERS: Figures 2, 3 and 4 were re-edited for better reading, including better clarity, larger font, more notes and Supplementary materials were also uploaded to the system along with the manuscript.

  1. In the materials and methods section, it is desirable to describe in detail the steps of the protocol by Ishida et al., it may be worth moving it to the Supplementry materials.

ANSWERS: The details of the protocol by Ishida et al were added in the Supplementary file 3.

  1. And it is better to describe in more detail the process of selecting controls CK1 and CK2. Were they also incubated on WLS-AS medium only without Аgrobacterium tumefaciens? After all, the composition of the medium can affect the results of your research.
    How many immature wheat embryos were used?
    Since you have a complex study with a large amount of data, it is better to describe the experiment itself in more detail.

ANSWERS: T1 and T2 were incubated on the WLS-AS medium with Agrobacterium tumefaciens. To control the experiment precisely enough and obtain reliable results, CK1 and CK2 were incubated on the WLS-AS medium without Agrobacterium tumefaciens.

In this experiment, a total of 120 POTS of wheat were planted in batches in the greenhouse, each pot could grow 6 plants and obtain 10 usable panicles. An average of 20 immature embryos were obtained from each panicle, and 300 immature embryos weigh about 1g. The transcriptome and metabolome used a total of about 30 g of embryos.

We have made detailed additions to the materials and methods to make readers better understand the article.

We tried our best to improve the manuscript and made some changes in the manuscript. These changes will not influence the content and framework of the paper. And here we did not list the changes but marked in red in revised paper.

We appreciate for Editors/Reviewers’ warm work earnestly and hope that the correction will meet with approval.

Once again, thank you very much for your comments and suggestions

Sincerely

Yajun Xi

30 April 2023
